# Vision Transformers Enable Fast and Robust Accelerated MRI

**Kang Lin**[1]                                                                    KA.LIN@TUM.DE

**Reinhard Heckel**[1,2]                                               REINHARD.HECKEL@TUM.DE

[1] *Department of Electrical and Computer Engineering, Technical University of Munich*

[2] *Department of Electrical and Computer Engineering, Rice University*

## Abstract

The Vision Transformer, when trained or pre-trained on datasets consisting of millions of images, gives excellent accuracy for image classification tasks and offers computational savings relative to convolutional neural networks. Motivated by potential accuracy gains and computational savings, we study Vision Transformers for accelerated magnetic resonance image reconstruction. We show that, when trained on the fastMRI dataset, a popular dataset for accelerated MRI only consisting of thousands of images, a Vision Transformer tailored to image reconstruction yields on par reconstruction accuracy with the U-net while enjoying higher throughput and less memory consumption. Furthermore, as Transformers are known to perform best with large-scale pre-training, but MRI data is costly to obtain, we propose a simple yet effective pre-training, which solely relies on big natural image datasets, such as ImageNet. We show that pre-training the Vision Transformer drastically improves training data efficiency for accelerated MRI, and increases robustness towards anatomy shifts. In the regime where only 100 MRI training images are available, the pre-trained Vision Transformer achieves significantly better image quality than pre-trained convolutional networks and the current state-of-the-art. Our code is available at https://github.com/MLI-lab/transformers_for_imaging.

**Keywords:** Accelerated MRI, Transformer, pre-training, image reconstruction

## 1. Introduction

Magnetic resonance imaging (MRI) is a medical imaging technique that provides excellent soft-tissue contrast, and is considered to be a safe diagnostic tool with the capability to reliably detect a wide range of diseases such as tumors, hemorrhage, and infections.

However, the data acquisition in MRI is inherently slow, leading to time-consuming examinations. This downside makes MRI particularly unsuitable for patients who struggle to remain still for longer periods of time (e.g., children) since even small movements increase the risk of image artifacts (Zaitsev et al., 2015).

To accelerate the examination process, MRI is usually accelerated by only collecting a few undersampled measurements. To enable reconstruction from few measurements, reconstruction algorithms need to incorporate prior knowledge about MRI images. Classically this is done without any training data, for example by assuming that the images are sparse in some basis (Lustig et al., 2007; Candes and Wakin, 2008).

Recently, deep learning methods have shown to outperform traditional methods in reconstruction quality and speed (Hammernik et al., 2018; Zbontar et al., 2019; Sriram et al., 2020; Darestani and Heckel, 2021; Jalal et al., 2021).

Current state-of-the-art deep learning based reconstruction methods all deploy convolutional neural networks (CNNs) as a core building block. The success of CNNs is often partially attributed to the inductive biases inherent in CNNs, allowing impressive data efficiency.

However, the *Vision Transformer* (ViT) (Dosovitskiy et al., 2020) — a convolution-free architecture with minimal inductive bias — has recently demonstrated superior performance over state-of-the-art CNNs in image classification when trained on millions of images. This suggests that the inductive bias in CNNs may restrict their performance if large amounts of data are available, and in this regime, a ViT may learn better features directly from the training data itself. While many recent works have studied the ViT, or self-attention, for image classification (Dosovitskiy et al., 2020; d'Ascoli et al., 2021; Xiao et al., 2021; Li et al., 2021; Touvron et al., 2021; Liu et al., 2021) and image processing (Chen et al., 2021; Liang et al., 2021; Wang et al., 2021), the number of works exploring self-attention for accelerated MRI remains sparse (Feng et al., 2021a,b; Korkmaz et al., 2022, 2021).

In this work, we study the ViT for accelerated MRI. Our findings are the following:

- Even when only trained on the 35k-70k training images from the fastMRI dataset, the ViT already performs on par or better than the U-net for accelerated MRI. The U-net is a strong CNN baseline deployed in present state-of-the-art reconstruction methods.

- The ViT benefits from almost $2\times$ higher throughput and less memory consumption when compared to the U-net for accelerated MRI.

- We can improve ViT's performance for accelerated MRI using pre-training on natural image datasets like ImageNet, which are readily available. The pre-trained ViT achieves competitive results with the state-of-the-art and outperforms pre-trained U-nets after fine-tuning on the fastMRI dataset. A pre-trained ViT is particulary interesting in the low-data regime: Even when only 100 MRI images for fine-tuning are available, the pre-trained ViT can still provide sharp and detailed reconstructions, and outperforms the U-net in this regime.

- We further show that pre-trained ViTs are also more robust towards anatomy shifts when compared to the current state-of-the-art convolutional neural networks.

## 1.1. Problem Formulation

During an accelerated MRI scan, electromagnetic waves are measured by several receiver coils. These measurements are typically referred to as k-space measurements, and are given by

$$\mathbf{y}_i = \mathbf{PFS}_i\mathbf{x}^* + \mathbf{z}_i \in \mathbb{C}^m, \quad i = 1, \dots, C. \tag{1}$$

Here, $\mathbf{x}^* \in \mathbb{C}^n$ is the unknown, vectorized image, $\mathbf{S}_i \in \mathbb{C}^{n \times n}$ are the sensitivity maps pertaining to the $C$ receiver coils (realized as a diagonal matrix), $\mathbf{F} \in \mathbb{C}^{n \times n}$ denotes the 2D discrete Fourier transformation, $\mathbf{P} \in \mathbb{R}^{m \times n}$ containing $m < n$ rows of an $n \times n$ identity matrix describes the undersampling operation, $\mathbf{z}_i \in \mathbb{C}^n$ models additive white Gaussian noise. For $C > 1$ many receiver coils the setup is referred to as *multi-coil* MRI, and for $C = 1$, it is referred to as *single-coil* MRI. Our goal is to reconstruct the image $\mathbf{x}^*$ from the undersampled measurements $\mathbf{y}_i$.

## 2. Related Work

In this work we study the Vision Transformer (ViT) for accelerated magnetic resonance imaging. Our work is motivated by the fact that *standard* Transformers scale well with the amount of training data for natural language processing and computer vision tasks (Devlin et al., 2019; Brown et al., 2020; Dosovitskiy et al., 2020).

Since the introduction of Vision Transformers and their revolutionary success in computer vision (Dosovitskiy et al., 2020), a number of recent works have proposed to utilize Transformers or self-attention mechanisms also in image reconstruction methods.

A number of works build attention mechanisms into networks that are U-shaped like the convolutional U-net (Ronneberger et al., 2015). Specifically, Wang et al. (2021), Ji et al. (2021), and Zamir et al. (2021) proposed U-shaped Transformer architectures, which use their own variants of efficient self-attention mechanisms, opening up the possibility to process high resolution images. Similarly, Liang et al. (2021) proposed an architecture based on the Swin Transformer (Liu et al., 2021), which performs self-attention locally in shifting windows, introducing locality biases that result in higher data-efficiency.

Most related to our approach is the work by Chen et al. (2021), which proposed a pre-trained backbone model based on the standard ViT for handling a variety of image restoration tasks simultaneously. The work showed that pre-trained ViTs are efficient models for image denoising and super-resolution when trained on sufficiently large datasets. Contrary to our architecture, however, Chen et al. (2021) still use convolutional layers in their architecture.

A few recent works have also explored self-attention for accelerated MRI reconstruction. Guo and Patel (2021) proposed the Texture Transformer module, which can be appended to existing architectures, such as the U-net to improve performance for MRI reconstruction. Feng et al. (2021a) proposed a new cross-attention module that can be combined with Transformers for multi-modal MRI reconstruction. Feng et al. (2021b) proposed the Task Transformer, a special Transformer architecture, for jointly performing MRI reconstruction and super-resolution. Korkmaz et al. (2022, 2021) use a Transformer architecture to perform MRI reconstruction in a deep image prior fashion (Ulyanov et al., 2018), eliminating the reliance on training data.

In our work, we use the original ViT from Dosovitskiy et al. (2020) with a straightforward adaption to image reconstruction, and also without any convolutions, to perform accelerated MRI as an end-to-end trained reconstruction method, like the U-net (Zbontar et al., 2019).

## 3. Image Reconstruction Transformer

In this section we describe the Transformer architecture that we consider, which is an adaption of the original Vision Transformer for image reconstruction tasks.

**Vision Transformer:** The Vision Transformer (ViT), proposed by Dosovitskiy et al. (2020), is an application of the original Transformer encoder (Vaswani et al., 2017) to image classification tasks. The Vision Transformer maps an input image to features as follows.

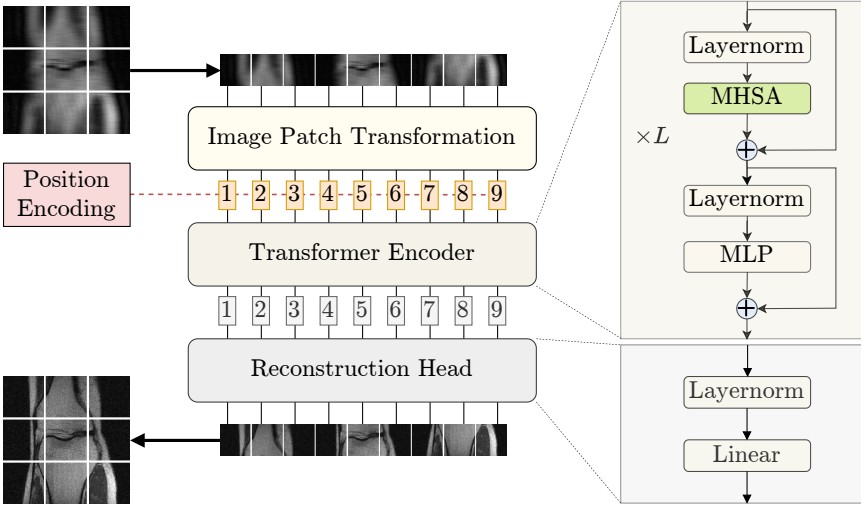

Figure 1: Vision Transformer for image reconstruction. As in the original ViT for classification, we linearly embed patches of the input image, and send them together with position embeddings into a standard Transformer encoder. At the output of the Transformer, we use a reconstruction head that maps the sequence elements back to a visual image.

First, the input image is spatially divided into a sequence of $N$ equally sized image patches. Then, a trainable linear transformation maps each of those image patches to a $d$-dimensional feature vector called *patch embedding*.

Since the Transformer encoder itself does not inherit any notion of positional information, $N$ learnable *position embeddings* are introduced. These position embeddings are $d$-dimensional vectors, which encode information about the absolute position of sequence elements. They are added to the patch embeddings prior to entering the Transformer encoder.

A learnable *classification token* is prepended to the sequence of patch embeddings. These $N + 1$ feature vectors serve as input to the Transformer encoder. The Transformer encoder consists of $L$ encoder layers, whereby each encoder layer contains a Multi-Head Self-Attention (MHSA) block, see Appendix A, and a two-layer MLP block that transforms each feature vector separately. Moreover, Layernorm (Ba et al., 2016) is applied before each block, and residual connections (He et al., 2016) after each block to help with training.

At the output of the Transformer encoder, only the output representation of the classification token is fed into a *classification head*, which returns the estimated class label of the input image. The classification head itself may be implemented by a multilayer perceptron (MLP) or a simple linear layer.

**Adapting the Vision Transformer to image reconstruction:** Figure 1 illustrates the ViT for image reconstruction. For this work, we adapted the original ViT (Dosovitskiy et al., 2020) to image reconstruction by performing two simple modifications.

First, we discard the classification token as it becomes redundant for image reconstruction. Second, we replace the classification head by a *reconstruction head* that maps the Transformer output back to a visual image.

The reconstruction head contains one Layernorm followed by a linear layer, which are shared across all the sequence elements. Hence, each sequence element in the feature space is mapped to a corresponding image patch in pixel space. The reconstructed image patches are then combined to a full-sized image.

Contrary to other approaches that apply special versions of Transformers in combination with convolutions (Liang et al., 2021; Wang et al., 2021; Ji et al., 2021; Zamir et al., 2021), our architecture is convolution-free and uses only the standard Transformer encoder. Our approach is therefore most similar to the Image Processing Transformer (Chen et al., 2021), which, like us, uses a standard Transformer encoder, however, has a convolutional image patch transformation and reconstruction head.

## 4. Experiments

We evaluate the performance of ViTs for accelerated MRI on the fastMRI dataset. We find that, when only trained on the fastMRI dataset, the ViT achieves on par performance with the U-net while benefiting from computational savings. We also propose a setup for pre-training ViTs on big natural image datasets, which, after fine-tuning on the fastMRI dataset, results in competitive performance with the current state-of-the-art.

### 4.1. Setup

We consider the reconstruction of knee and brain MRIs in a 4-fold accelerated MRI setting, where we follow the undersampling procedure described by Zbontar et al. (2019, Sec. 4.9): The central k-space region (or with a random offset for brain scans) is first fully sampled containing 8% of all vertical k-space lines. To achieve the desired acceleration factor, the remaining k-space lines are either sampled uniformly at random for knee scans or equidistantly for brain scans.

**Datasets.** We use the fastMRI dataset (Zbontar et al., 2019) since its the largest public MRI dataset. This dataset is comprised of a collection of knee and brain MRIs. The knee dataset holds 35k slices for training and 7k for validation whereas 70k slices for training and 21k for validation are contained in the brain dataset.

**Model variants.** We experiment with three different ViT variants containing 8M, 32M, or 60M parameters, using a patch size of 10. They are denoted ViT-S, ViT-M, and ViT-L.

As a standard CNN baseline, we consider U-nets of different sizes. Our U-net variants have 4 down-sampling layers and 32, 64, or 128 channels in the first layer, which correspond to roughly 8M, 31M, and 124M model parameters, respectively, denoted by U-net-S, U-net-M, and U-net-L. Note, that the U-net-L has twice as many parameters as ViT-L.

We also compare the ViT to the End-to-end VarNet (Sriram et al., 2020) in Sec. 4.3. The End-to-end VarNet is convolution based and gives current state-of-the-art performance for accelerated MRI.

Given the undersampled k-space coil measurements $\mathbf{y}_i$, the ViT and U-net models take as input the root-sum-of-square reconstruction of the zero-filled coil images, where we ignore the estimation of sensitivity maps, and output a real-valued reconstructed image. Thus, the model input is given by $\sqrt{\sum_{i=1}^{C} |\mathbf{F}^{\mathsf{H}}\mathbf{P}^{\mathsf{H}}\mathbf{y}_i|^2}$, where the square and the root operator are to

Table 1: Reconstruction SSIM of our models when trained on the fastMRI dataset for 4-fold accelerated multi-coil (MC) and single-coil (SC) MRI, and their empirical computational costs during inference measured by throughput and maximal possible batch size. Best results are reported in bold, second best are underlined. ViT-L outperforms all other methods, even the U-net-L which has twice as many parameters. In addition, the ViTs have a higher throughput, and are thus computationally cheaper than the corresponding U-net versions.

| | ViT-L | ViT-M | ViT-S | U-net-L | U-net-M | U-net-S |
|---|---|---|---|---|---|---|
| MC-Knee | $\mathbf{0.908}_{\pm 0.118}$ | $\underline{0.907}_{\pm 0.118}$ | $0.903_{\pm 0.118}$ | $\underline{0.907}_{\pm 0.118}$ | $0.906_{\pm 0.118}$ | $0.905_{\pm 0.118}$ |
| SC-Knee | $\mathbf{0.744}_{\pm 0.250}$ | $\mathbf{0.744}_{\pm 0.249}$ | $0.740_{\pm 0.248}$ | $\mathbf{0.744}_{\pm 0.248}$ | $\underline{0.743}_{\pm 0.248}$ | $0.742_{\pm 0.248}$ |
| SC-Brain | $\mathbf{0.828}_{\pm 0.148}$ | $0.826_{\pm 0.148}$ | $0.823_{\pm 0.148}$ | $\underline{0.827}_{\pm 0.148}$ | $0.826_{\pm 0.148}$ | $0.825_{\pm 0.148}$ |
| Throughput | 97.4 img/s | 183.32 img/s | 442.96 img/s | 51.8 img/s | 153.35 img/s | 331 img/s |
| Batch size | 272 | 380 | 440 | 145 | 240 | 512 |

be interpreted entry-wise, and $|\cdot|$ takes the absolute value entry-wise. The ViT and U-net only take the information about the MRI forward model into account in the model input by using the root-sum-of-squares algorithm.

Contrary, the End-to-end VarNet, uses information about the MRI forward model (1) throughout the network.

**Training and validation.** We train all models with the objective to maximize the structural similarity index measure (SSIM) between model output and the ground-truth image. During training, we randomly sample a different undersampling mask for each training instance independently. During validation, each volume is assigned a different mask that is used for all slices within the volume. We report reconstruction accuracy on fastMRI's validation set. For further details regarding training, see Appendix B.2.

## 4.2. Comparing ViT to U-net

We start by comparing our ViT models to the U-net baselines. The U-net is commonly used as a standard baseline for the fastMRI dataset. We compare to the U-net, as it gives the best performance among models that map the root-sum-of-squares reconstruction directly to a clean image, just like the ViT does.

We later also compare to the state-of-the-art, End-to-end VarNet, which incorporates data-consistency steps and other elements in the network architecture.

Table 1, which contains the image reconstruction performance of our models when trained on the fastMRI dataset, shows that a large ViT outperforms the best U-net.

Furthermore, when comparing the empirical computational costs during inference, the results in the table show that all our ViT models beat their similar sized U-net counterparts in terms of throughput. Looking at the memory consumption, we notice that the two larger ViT variants (ViT-M and ViT-L) can operate on significantly larger batch sizes than the two larger U-net variants (U-net-M and U-net-L), indicating a clear advantage in memory efficiency. For further details on how we measured the computational costs, see Appendix B.3.

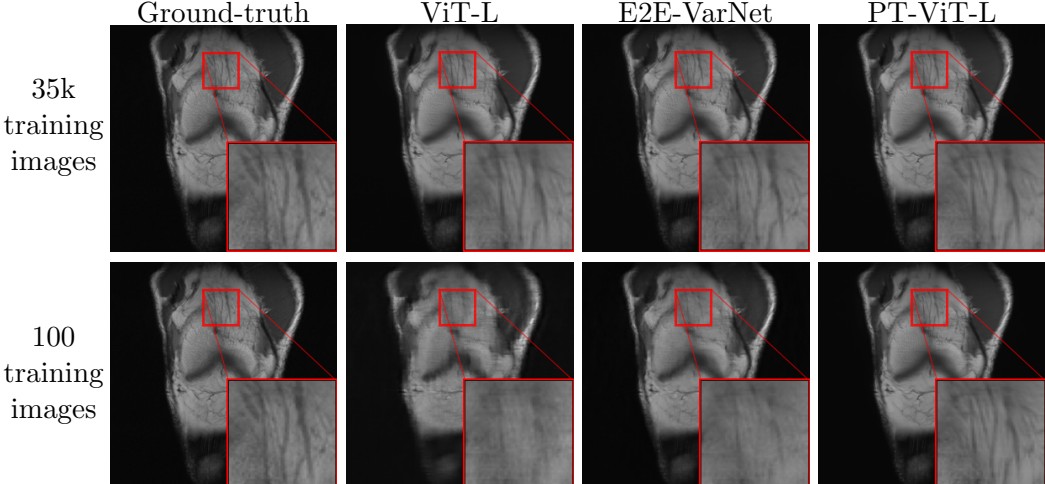

Figure 2: Example reconstructions for the 4-fold multi-coil knee setup. Models were trained on 35k data (**top**) and 100 data (**bottom**). The pre-trained ViT, PT-ViT-L, yields a detailed reconstruction even when just fine-tuned on 100 images.

### 4.3. Improving ViTs for MRI with Pre-training on Natural Images

Since MRI datasets are costly to obtain, and Transformers are known to perform best with large-scale training or large scale pre-training (Devlin et al., 2019; Brown et al., 2020; Dosovitskiy et al., 2020), we investigate to what extent pre-training on ImageNet (Russakovsky et al., 2015) improves the image reconstruction performance of ViTs for accelerated MRI.

For pre-training on ImageNet, we construct training data by feeding the ImageNet images through the forward model (1), where we randomly vary the acceleration factor. Further details for this setup are provided in Appendix B.2.

We pre-train a ViT-L, denoted PT-ViT-L, on ImageNet and fine-tune it on fastMRI's multi-coil knee dataset. Figure 2 shows example reconstructions that illustrate the benefits of pre-training. We observe that PT-ViT-L yields similar image quality to End-to-end VarNet when 35k training images (whole dataset) are available.

PT-ViT-L really shines in the regime where few training images are available: Even when only 100 training images are available, PT-ViT-L still yields a sharp and detailed reconstruction, demonstrating promising performance for the low-data regime.

Figure 3 provides a more comprehensive overview for the data requirements of our models. The figure depicts the reconstruction SSIM of our models when trained or fine-tuned on 100, 1k, or 35k knee images. Here, we additionally compare to a pre-trained U-net-L, denoted PT-U-net-L. We observe that PT-ViT-L consistently outperforms PT-U-net-L and their non-pre-trained counterparts. Indeed, in the regime where only 100 MRI training images are available, PT-ViT-L also outperforms the End-to-end VarNet, and reaches even on par performance with the non-pre-trained ViT-L for 35k training data. However, as dataset size increases, we observe diminishing returns, resulting in End-to-end VarNet slightly outperforming the PT-ViT-L at 35k data. Pre-training the U-net was not nearly as effective as pre-training the ViT, resulting only in slight performance gains in the low data regime.

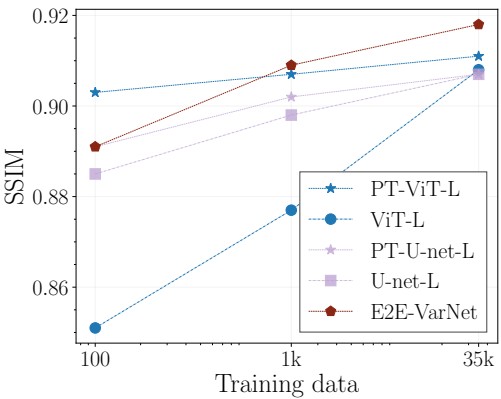

Figure 3: Performance of pre-training on ImageNet for the ViT and U-net with subsequent fine-tuning on the fastMRI multi-coil knee dataset. Pre-training the ViT results in a significant boost in reconstruction accuracy when the amount of fine-tuning data is low: when fine-tuned on only 100 images, the pre-trained ViT significanlty outperforms pre-trained and non-pretrained competitors.

Table 2: Performance under anatomy shift. Models are trained/fine-tuned on the entire fastMRI's knee dataset and then evaluated on the brain dataset. Best results are reported in bold, second best are underlined.

| Model | Knee SSIM (Trained on) | Brain SSIM (Not trained on) |
|---|---|---|
| ViT-L | $0.908 \pm 0.12$ | $0.916 \pm 0.08$ |
| PT-ViT-L | $\underline{0.911} \pm 0.12$ | $\mathbf{0.926} \pm 0.07$ |
| E2E-VarNet | $\mathbf{0.918} \pm 0.12$ | $\underline{0.923} \pm 0.09$ |

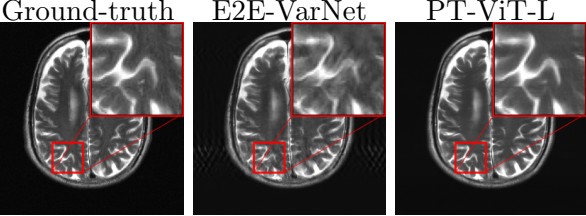

Figure 4: Anatomy shift example. Clear artifact are visible in the E2E-VarNet reconstruction, while PT-ViT-L gives a clean reconstruction.

Moreover, pre-training the ViT enhances robustness towards anatomy shifts, as suggested by the results in Table 2. While End-to-end VarNet gives better in-distribution results than PT-ViT-L, we observe under the anatomy shift that PT-ViT-L performs better than End-to-end VarNet, as indicated by higher SSIM and tighter confidence intervals. Looking at the example reconstructions provided in Fig. 4, we see clear artifacts in the End-to-end VarNet reconstruction under the anatomy shift that are not present in the PT-ViT-L reconstruction.

## 5. Conclusion

We investigated the application of convolution-free Vision Transformers for accelerated MR image reconstruction. If a sufficiently large Vision Transformer is trained on the fastMRI dataset, it slightly outperforms the U-net in terms of image quality, and notably outperforms the U-net in memory and computational performance. Moreover, we showed that a Vision Transformer pre-trained on ImageNet can compete with the current state-of-the-art convolutional method after fine-tuning on the fastMRI dataset. In particular we find that a ViT is very effective in the low-data regime which is common in MRI due to many different acquisition modes: Even when only fine-tuned on 100 MRI images, a pre-trained Vision Transformer yields sharp and detailed reconstructions, showing that the ViT is very suitable for low-data regimes in accelerated MRI.

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

Table 3: Hyperparameters of our ViT variants.

| Model | Parameters | Patch size | Layers $L$ | Dimension $d$ | MLP width | Heads $H$ |
|-------|-----------|-----------|-----------|--------------|-----------|-----------|
| ViT-S | 8M | 10 | 4 | 396 | 1584 | 9 |
| ViT-M | 32M | 10 | 8 | 576 | 2304 | 9 |
| ViT-L | 60M | 10 | 10 | 704 | 2816 | 16 |

## Appendix A. Multi-Head Self-Attention

Vaswani et al. (2017) introduced Multi-Head Self-Attention (MHSA) as the key component for their sequence processing neural network, called *Transformer*, which has enjoyed great success in natural language processing tasks (Devlin et al., 2019; Brown et al., 2020). In the following, we revisit MHSA in more detail.

Given a sequence of $N$ elements $\boldsymbol{\xi}_1, \ldots, \boldsymbol{\xi}_N \in \mathbb{R}^d$ at the input of a MHSA block, each element $\boldsymbol{\xi}_j$ is mapped to $H$ *queries* $\mathbf{q}_i \in \mathbb{R}^{d_H}$, *keys* $\mathbf{k}_i \in \mathbb{R}^{d_H}$ and *values* $\mathbf{v}_i \in \mathbb{R}^{d_H}$, where $i = 1, 2, \ldots, H$, $H$ denotes the total number of 'heads', and $d_H = d/H$ is the dimensionality of one head. This map is typically a trainable linear transformation.

In each head $i$, the self-attention mechanism, denoted by Attention: $\mathbb{R}^{d_H} \times \mathbb{R}^{d_H \times N} \times \mathbb{R}^{d_H \times N} \to \mathbb{R}^{d_H}$, computes for each element $\boldsymbol{\xi}_j$ a weighted average of the $N$ values, where the weights are computed by correlating its query with all $N$ keys. Thus, self-attention has the form

$$\text{Attention}\,(\mathbf{q}_i, \mathbf{K}_i, \mathbf{V}_i) = \mathbf{V}_i \, \text{softmax}\left(\frac{\mathbf{K}_i^{\mathsf{T}} \mathbf{q}_i}{\sqrt{d_H}}\right), \tag{2}$$

where the matrices $\mathbf{K}_i = \left[\mathbf{k}_i^{(1)}, \ldots, \mathbf{k}_i^{(N)}\right]$ and $\mathbf{V}_i = \left[\mathbf{v}_i^{(1)}, \ldots, \mathbf{v}_i^{(N)}\right]$ concatenate the keys and values, respectively.

Lastly, we concatenate the attention outputs of each head to form a single vector in $\mathbb{R}^d$ again, and transform this vector using a learnable weight matrix $\mathbf{W}^{\text{out}} \in \mathbb{R}^{d \times d}$, i.e.,

$$\boldsymbol{\xi}_j^{\text{out}} = \mathbf{W}^{\text{out}} \begin{bmatrix} \text{Attention}\,(\mathbf{q}_1, \mathbf{K}_1, \mathbf{V}_1) \\ \vdots \\ \text{Attention}\,(\mathbf{q}_H, \mathbf{K}_H, \mathbf{V}_H) \end{bmatrix}, \tag{3}$$

where $\boldsymbol{\xi}_j^{\text{out}}$ denotes the output representation of the input element $\boldsymbol{\xi}_j$ after leaving the MHSA block.

## Appendix B. Details of Experimental Setup

### B.1. Hyperparameters

Table 3 provides the hyperparameters of our ViT models from Sec 4.1.

### B.2. Training

We train all models with the structural similarity index measure (SSIM) loss between model output and target image and with the Adam optimizer (Kingma and Ba, 2015) with hyperparameters $\beta_1 = 0.9$, $\beta_2 = 0.999$, and no weight decay. Additionally, we clip the gradients

Ground-truth       PT-ViT-L, 100       E2E-VarNet, 35k       PT-ViT-L, 35k

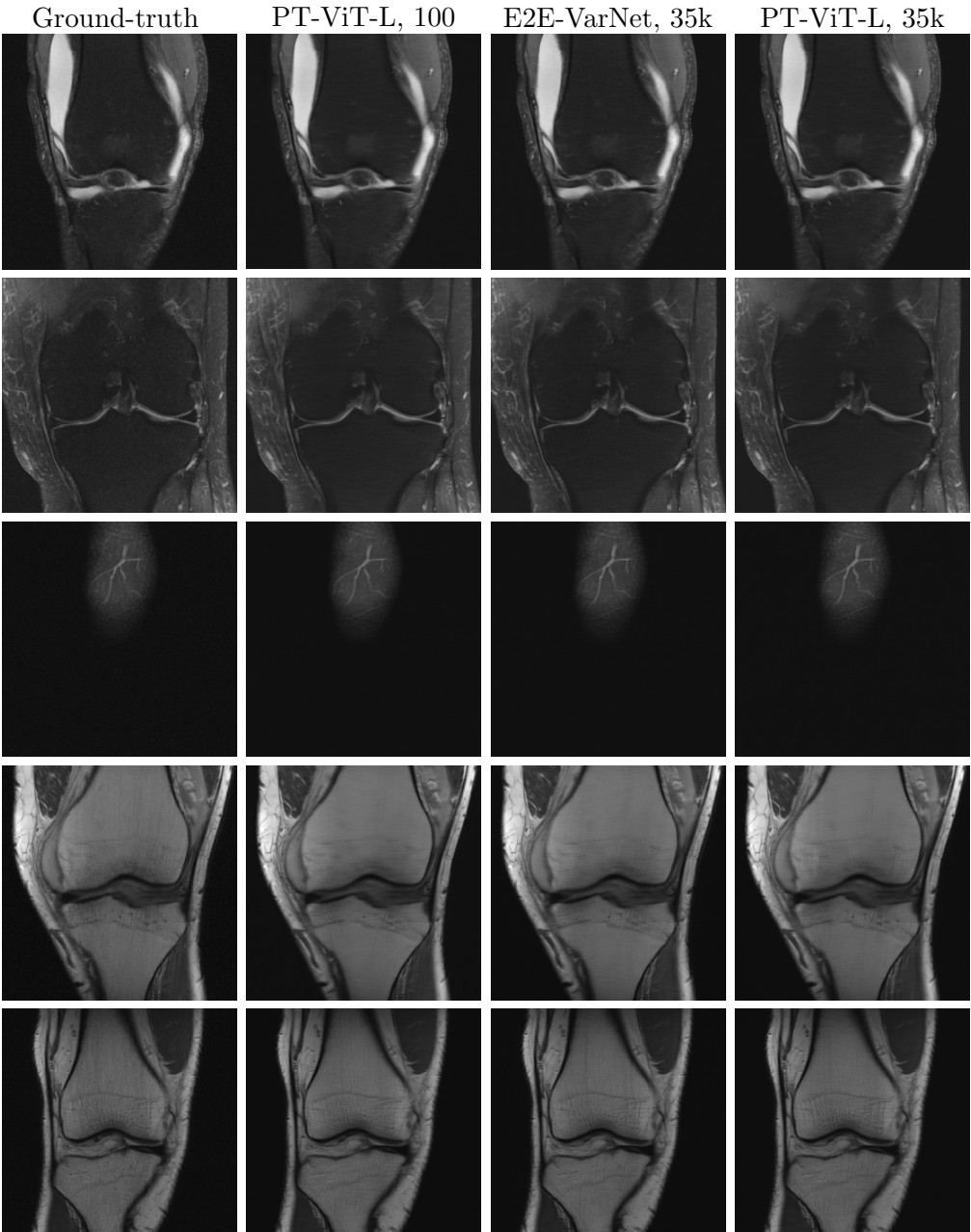

Figure 5: Random selection of reconstructions for the 4-fold accelerated multi-coil knee set-up. The number behind the model names denotes the amount of data used for training or fine-tuning. The pre-trained ViT can provide competitive image quality with End-to-end VarNet.

at a global $\ell_1$-norm of 1 to help with training stability. The models are validated every 5 epochs, of which the best validation score is reported.

Table 4: Performance under anatomy shift. Models are trained or fine-tuned on the entire fastMRI's knee dataset and then evaluated on the brain dataset. Best results are reported in bold. The pre-trained ViT consistently achieves better results under the anatomy shift.

| Train. data | Model | Knee SSIM (Trained on) | Brain SSIM (Not trained on) |
|---|---|---|---|
| 35k | PT-ViT-L | $0.911 \pm 0.12$ | $\mathbf{0.926} \pm 0.07$ |
|  | E2E-VarNet | $\mathbf{0.918} \pm 0.12$ | $0.923 \pm 0.09$ |
| 1k | PT-ViT-L | $0.907 \pm 0.12$ | $\mathbf{0.926} \pm 0.07$ |
|  | E2E-VarNet | $\mathbf{0.908} \pm 0.12$ | $0.905 \pm 0.1$ |
| 100 | PT-ViT-L | $\mathbf{0.904} \pm 0.12$ | $\mathbf{0.922} \pm 0.07$ |
|  | E2E-VarNet | $0.890 \pm 0.12$ | $0.895 \pm 0.1$ |

Table 5: Reconstruction performance on 8-fold accelerated multi-coil knee MRIs. Best results are reported in bold, second best are underlined.

|  | PT-U-net-L | PT-ViT-L | E2E-VarNet |
|---|---|---|---|
| SSIM | $0.875 \pm 0.138$ | $\underline{0.881} \pm 0.135$ | $\mathbf{0.887} \pm 0.136$ |

| Ground-truth | PT-U-net-L | PT-ViT-L | E2E-VarNet |
|---|---|---|---|

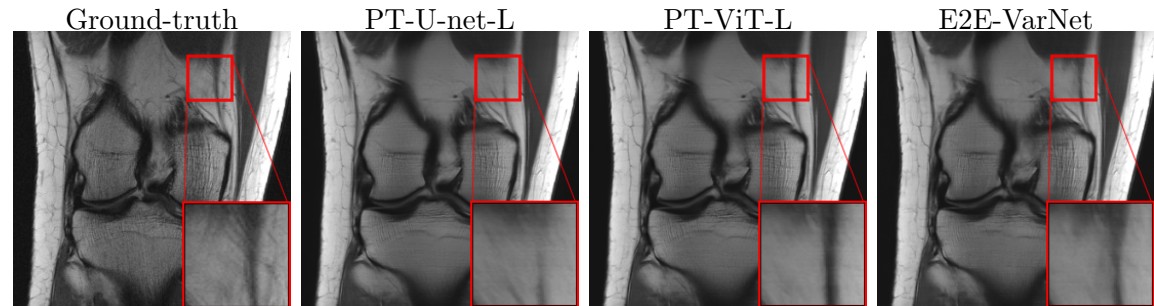

Figure 6: Example reconstructions for the 8-fold accelerated multi-coil knee setup. PT-U-net-L fails to reconstruct the highlighted area, which PT-ViT-L and End-to-end VarNet successfully reconstructed. However, End-to-end VarNet gives a slightly more accurate reconstruction.

**Training on fastMRI dataset.** For training on the fastMRI dataset, we use a mini-batch size of 1, and train all models for 40 epochs using a linear learning rate warm-up and decay. We use 4 warm-up epochs for all models. The base learning rates for ViT-L, U-net-M, and U-net-S are 0.0005, while U-net-L, ViT-M, and ViT-S use 0.0003. The End-to-end VarNet uses a base learning rate of 0.003. In order for the ViT to handle varying image sizes, we first initialized the trainable position embeddings according to the average image size estimated on the training dataset, and then bi-linearly interpolated between the embedding values.

**Pre-training and fine-tuning.** For pre-training on ImageNet, we first gray-scale each image and resize them such that the smaller edge has size 320 and proportionally adjusted the longer edge. Then, we randomly crop out an image patch of size $272 \times 272$, perform a random flip and rotation by $\pm 90$ degrees, and use this augmented image as ground-truth. We take this ground-truth to construct undersampled Fourier measurements by following the forward model (1), where we randomly vary the acceleration factor between 2 and 10. We use a batch-size of 32 and train 10 epochs with a learning rate of 0.0005, then 4 epochs with a learning rate of 0.0003, and another 2 epochs with a learning rate of 0.0001. For fine-tuning, we re-sample the position embeddings using bi-linear interpolation to fit the average image size on the fastMRI dataset. We fine-tune for 30 epochs with a batch size of 1 and use a linear learning rate schedule. We use 3 warm-up epochs and a base learning rate of 0.0001.

### B.3. Empirical Computational Costs

To measure the empirical computational costs of our models, we deploy our models on a single NVIDIA RTX A6000 GPU, and measure their throughput as well as their memory usage by fitting the largest possible batch size on the device. The models operate on gray scale images of size $320 \times 320$ as this image size is typical for our experiments.

## Appendix C. Additional Results and Insights

### C.1. Additional Example Reconstructions

We provide a set of random reconstructions in Fig. 5 for the 4-fold accelerated multi-coil knee setup. We observe that the pre-trained ViT, PT-ViT-L, can consistently provide image quality competitive with the state-of-the-art, End-to-end VarNet, even when only fine-tuned on 100 training images.

### C.2. Additional Anatomy Shift Experiments

In the following, we extend our anatomy shift results from Sec 4.3 by evaluating the anatomy shift performance of the PT-ViT-L and End-to-end VarNet as a function of fine-tuning/training data used.

Table 4 shows the results. We fine-tune/train PT-ViT-L and End-to-end VarNet on either 100, 1k, or 35k training images from fastMRI's multi-coil knee dataset for 4-fold acceleration, and then evaluate on fastMRI's multi-coil brain dataset. The pre-trained ViT consistently achieves better results under the anatomy shift, as indicated by higher SSIM and tighter confidence intervals.

### C.3. Results for 8-fold Accelerated Multi-Coil Knee MRI

We also train and evaluate our models for the 8-fold accelerated multi-coil knee MRI task, as it is a common benchmark task alongside the 4-fold acceleration setup (Zbontar et al., 2019; Sriram et al., 2020). For this setup, we fine-tune the pre-trained U-net-L (PT-U-net-L), the pre-trained ViT-L (PT-ViT-L), and trained the End-to-end VarNet.

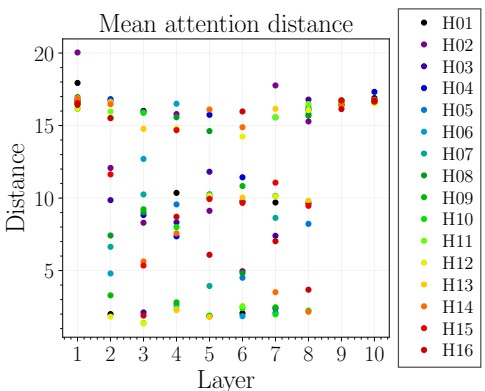

Figure 7: Mean attention distance of each attention head for each layer averaged over 500 random images. The attention distance is calculated by weighing the distances between queries and keys with their attention weights.

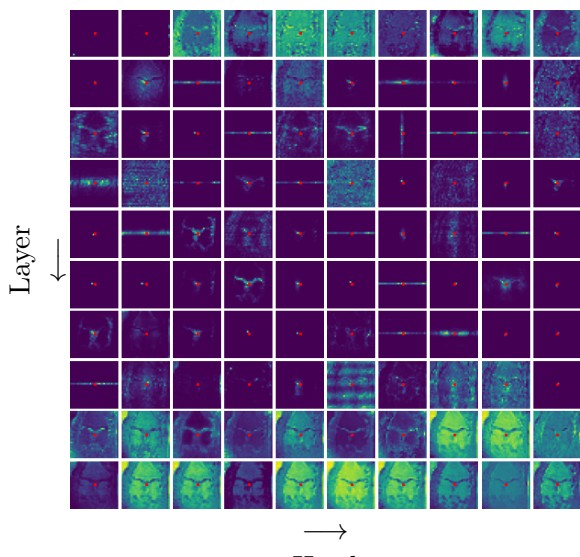

Figure 8: Attention maps for an arbitrarily chosen input image. Depicted are the attention maps of one query (**red dot**) for the first 10 attention heads of each layer.

Table 5 shows the reconstruction SSIM of our models on the fastMRI validation set. We observe that PT-ViT-L performs better than PT-U-net-L, however, still slightly worse than End-to-end VarNet. Figure 6 provides example reconstruction, where we notice that PT-U-net-L fails to reconstruct the highlighted area, which, however, is successfully reconstructed by PT-ViT-L and End-to-end VarNet.

## C.4. What Does a ViT for MRI Reconstruction Attend To?

In the following we inspect the learned attention weights of a fine-tuned PT-ViT-L.

Figure 7 plots the mean attention distance of each head in each layer, which is analogous to the receptive field of a convolutional network. The attention distance for one input is calculated by weighing the distances between queries and keys with their attention weights (d'Ascoli et al., 2021). From the figure, we observe that all the attention heads in the first layer consistently apply global attention, which can be interpreted as global information exchange. This behavior changes abruptly starting from layer 2 all the way up to layer 8, where we see an almost uniform mix of global and a local attention. Finally, in the last two layers the attention behavior changes back abruptly to consistent global attention.

Figure 8 shows the attention maps of our model for one example input and one query element. Indeed, we can observe attention behavior as previously discussed. Interesting, however, are the attention maps in the middle layers, where the query element appears to mostly attend to keys that are positioned in the same horizontal line as the query. We suspect that this behavior is caused by our considered undersampling procedure (Sec. 4.1), which introduces blurring along the horizontal axis of an image.

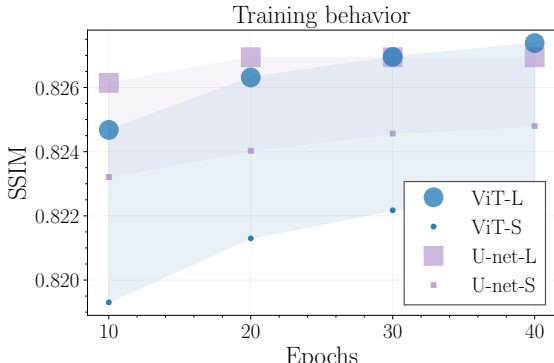

Figure 9: Reconstruction SSIM of our models on the single-coil brain dataset as a function of trained epochs. Training large ViT models tends to be more epoch efficient than training U-nets when given enough data.

We notice that the attention behavior of a ViT for image reconstruction is quite different from a ViT trained for classification (Dosovitskiy et al., 2020). A ViT for image classification typically start with a widely spread mix of global and local attention in the first layer, which monotonically converges to consistent global attention as we move to deeper layers.

### C.5. ViT Training Behavior

With reduced inductive bias, one might wonder how much training time the ViT needs in order to reach reasonable performance. In the following we inspect the training behavior of our models. Figure 9 depicts validation scores of our largest and smallest model variants after taking certain numbers of gradient steps. The models were trained on the single-coil brain dataset for 40 epochs with the same mini-batch size.

Contributed to their inductive bias, the U-net models attain high validation scores already early on during training but experience saturating returns after 50% of total training time, whereas for ViT the validation score starts at a lower point but seems to improve in a more steady fashion. Interestingly, even with drastically reduced inductive bias, the ViT-L enters the performance regime of the U-net after only 25% of total training time and overtakes the U-net-L after 75% of total training time.

For both the U-net and the ViT we notice that larger models need fewer epochs to reach higher validation scores. However, the ViT appears to benefit more effectively from increasing parameter counts than the U-net, since not only does our largest ViT model have significantly fewer parameters than our largest U-net model, but it also experiences a bigger performance gain when switching from the smallest variant.

These observations indicate that the ViT enjoys higher training efficiency than the U-net in high parameter and data regimes.

### C.6. Linear Transformer

For a regular Transformer, both computational time and space scale quadratically with the sequence length due to the computation of the self-attention matrix (2). For image reconstruction, this quadratic bottleneck quickly becomes prohibitively expensive with growing

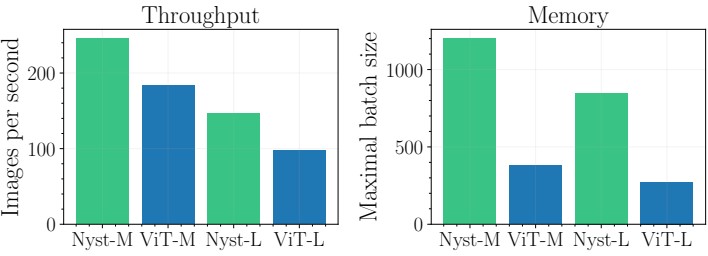

Table 6: Reconstruction accuracy of Nyströmformer variants for the 4-fold single-coil setup.

| Task | Model | SSIM |
|------|-------|------|
| Knee | Nyst-M | 0.7427 |
|      | ViT-M | 0.7438 |
| Brain | Nyst-L | 0.8274 |
|       | ViT-L | 0.8274 |

Figure 10: Empirical computational costs of Nyströmformer compared to ViT during inference when operating on gray scale images of size $320 \times 320$. The models were deployed on a single NVIDIA RTX A6000 GPU. **Left:** Throughput measured in images per second. **Right:** Largest possible batch size to fit on device. Note that the Nyströmformer is a linear Transformer and thus time and space complexity only scales linearly with increasing image size (as opposed to quadratic scaling in the case of ViT).

image size and limited computational resources. To mitigate the scaling issue, *linear* Transformers have been proposed, e.g., (Wang et al., 2020; Choromanski et al., 2020; Xiong et al., 2021), which allow linear time and space complexity by approximating the self-attention mechanism.

We found the *Nyströmformer* (Xiong et al., 2021) to be a promising linear Transformer variant for image reconstruction in accelerated MRI, as it reaches on par reconstruction accuracy with the regular ViT.

Table 6 shows the reconstruction performance on the 4-fold accelerated single-coil knee or brain MRI reconstruction task of a 60M and 32M parameter Nyströmformer, denoted Nyst-L and Nyst-M, respectively. Both linear Transformers have the same hyperparameter configurations as our regular ViT variants, and are also trained in the same fashion. We see that the Nyströmformer reaches comparable SSIM to the regular ViT in both reconstruction tasks while being much more computationally efficient than the regular ViT, especially in memory consumption, as shown in Fig. 10.

## C.7. Impact of Patch Size

Computational time and space of a ViT scales quadratically with number of sequence elements. Therefore, increasing the patch size is of computational interest as larger patches lead to a smaller sequence length and thus a drastic decrease in time and memory consumption.

In this section, we investigate how a larger patch size impacts the reconstruction performance of ViTs. In all our previous experiments we used a patch size of 10. For the experiments in this section, we took our previous models and only changed the patch size from 10 to 16. Additionally, we introduce a new ViT variant with 152M parameters (12 layers, 16 heads with $d_H = 64$), denoted ViT-XL/16, that uses patch size 16. To distinguish

Table 7: Reconstruction accuracy of ViTs with different patch sizes on 4-fold single-coil knee MRI reconstruction task.

| Model | Patch size | SSIM |
|-------|-----------|------|
| ViT-S | 10 | 0.7402 |
|       | 16 | 0.7375 |
| ViT-M | 10 | 0.7438 |
|       | 16 | 0.7408 |

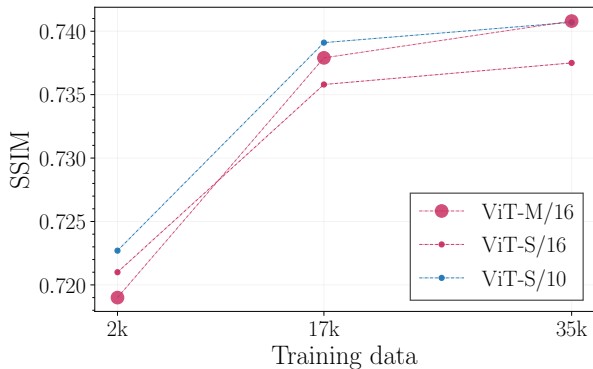

Figure 11: Data and parameters scaling of a patch size 16 ViT. In order for a larger patch size model to reach the same SSIM as a smaller patch size model, more parameters and data are needed.

between models in this section, we append ViT model names with '/16' or '/10' to refer to patch size 16 or patch size 10 models, respectively. We find that, given the same amount of data, the patch size 16 models yield slightly worse reconstruction accuracy.

**Reconstruction accuracy.** Table 7 compares the reconstruction accuracy (SSIM) of patch size 16 models to the patch size 10 models on the 4-fold single-coil knee MRI task. We see that the SSIM for the patch size 16 models dropped by a slight amount. However, we notice that the accuracy gap between the larger patch size model and the smaller patch size model may be compensated by scaling up the amount of data and the parameter count, as for example demonstrated in Fig. 11. Here, we observe that as the amount of data increases the SSIM difference between the smaller patch 10 model, ViT-S/10, and the larger patch 16 model, ViT-M/16, decreases. However, if we only increase the patch size without changing the model size, then the accuracy gap appears to continue as indicated by the accuracy curve of ViT-S/16.

**Computational resources.** Although the patch size 16 models need more parameters (and more data) than the patch size 10 models to perform equally well, they still have a clear computational advantage as shown in Table 8. We observe that the patch size 16 models, though having more parameters, still provide a significant throughput and especially memory advantage over the patch size 10 models. Setup is described in Appendix B.3.

**ImageNet pre-training.** Finally, when utilizing pre-training on ImageNet as described in 4.3, it is also possible for a patch size 16 model to perform on par with a U-net. Figure 12 shows the effect of pre-training and fine-tuning the ViT-XL/16 on the 4-fold single-coil brain MRI dataset. The pre-trained model is denoted by PT-ViT-XL/16. We observe that PT-ViT-XL/16 experiences a drastic performance gain over the non-pre-trained ViT-XL/16, while yielding on par results with U-net-L. Note that PT-ViT-XL/16 has an enormous speed and memory advantage over the U-net-L.

Table 8: Comparison of throughput and maximal batch-size between patch size 16 and patch size 10 models.

| Model | Parameters | Throughput | Batch size |
|-------|-----------|-----------|-----------|
| ViT-S/10 | 8M | 442 img/s | 440 |
| ViT-M/16 | 32M | 635 img/s | 2100 |
| ViT-L/10 | 60M | 97 img/s | 272 |
| ViT-XL/16 | 152M | 207 img/s | 1200 |

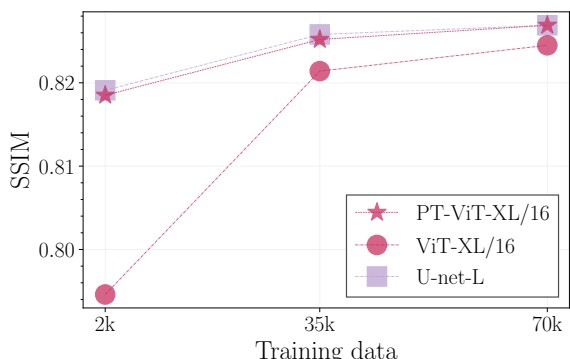

Figure 12: Impact of ImageNet pre-training on the patch size 16 ViT-XL with subsequent fine-tuning on the fastMRI single-coil brain dataset. Pre-training the ViT results in a significant boost in reconstruction performance, especially when the amount of fine-tuning data is low. The pre-trained patch size 16 ViT reaches on par accuracy with the U-net-L.

## C.8. Impact of the Loss Function

In this section we discuss how different choices of training objectives (loss functions) can impact model performance. We compare two training objectives: (i) minimizing the L1-loss, which we now refer to as 'L1-loss training', and (ii) maximizing the SSIM, which we now refer to as 'SSIM training'. Intuitively, L1-loss training should yield higher peak signal-to-noise ratio (PSNR) than SSIM training, while SSIM training should yield higher SSIM scores than L1-loss training.

Table 9 depicts our results for a ViT-M with either L1-loss training or SSIM training. SSIM training indeed results in significantly better evaluation SSIM than L1-loss training. However, L1-loss training does not provide much better PSNR than SSIM training, and in fact only reaches on par PSNR with SSIM training. Moreover, we also noticed during the experiments that SSIM training gives much better PSNR early on during the training process than L1-loss training.

These observations suggest that using a perceptual metric, such as SSIM, as loss function for training ViTs for image reconstruction might be more advantageous than choosing a pixel-wise loss function such as the L1-loss.

Table 9: Comparison of performance of a ViT-M on 4-fold accelerated single-coil knee dataset, when using L1-Loss or SSIM as training objective. Training on maximizing SSIM not only yields better SSIM at inference than L1-Loss training but also yields on-par PSNR with L1-Loss training.

| Training Objective | PSNR [dB] (evaluated) | SSIM (evaluated) |
|---|---|---|
| L1-Loss | $32.28 \pm 7.55$ | $0.726 \pm 0.267$ |
| SSIM | $32.22 \pm 7.99$ | $0.744 \pm 0.249$ |

Table 10: Results for ViT-VarNets on the 4-fold accelerated single-coil knee task. Using the ViT in a VarNet fashion hurts performance rather than improving it: ViT-S has 8M parameters and performs better than any of the ViT-VN variants which each have about 32M parameters.

| | ViT-VN-8/4M | ViT-VN-4/8M | ViT-VN-2/16M | ViT-S |
|---|---|---|---|---|
| SSIM | $0.725 \pm 0.249$ | $0.728 \pm 0.253$ | $0.731 \pm 0.251$ | $0.736 \pm 0.247$ |

## C.9. Data-consistency in ViTs

In all previous sections, we train the ViT to directly map a coarse root-sum-of-squares reconstruction to the ground-truth magnitude image, without incorporating data consistency with known k-space data. However, networks that incorporate data consistency steps typically give a boost in performance. In this section, we therefore experiment with data consistency techniques for ViTs.

**Replacing U-nets with ViTs in the VarNet.** The End-to-end VarNet (Sriram et al., 2020) yields state-of-the-art image reconstruction performance for deep learning based accelerated MRI, by combining data consistency steps with U-nets. In the following, we experiment with replacing U-nets with ViTs in the VarNet.

We train 3 ViT-VarNet variants with 32M parameters: (i) ViT-VN-8/4M, which has 8 cascades of 4M parameters sized ViTs, (ii) ViT-VN-4/8M, which has 4 cascades of 8M parameters sized ViTs, and (iii) ViT-VN-2/16M, which has 2 cascades of 16M parameters sized ViTs. We further note that we use a patch size of 16 for all ViTs used in this section. Moreover, for simplicity, we apply the End-to-end VarNet to the single-coil setup, which omits the need for sensitivity maps estimation networks.

Table 10 provides our results on the 4-fold accelerated single-coil knee dataset. We observe that, first, having fewer cascades with larger ViTs yields better performance than having more cascades but with smaller ViTs, and second, none of the ViT-VarNet variants outperform a simple ViT-S (only 8M parameters) that does not rely on any data consistency. Thus we did not identify a benefit by directly substituting Unets with ViTs in a VarNet, however there might be other options to use data consistency steps together with ViTs that yield a benefit.

**Data consistency at inference.** Other than using ViTs in a VarNet, we can also apply data consistency in the single-coil setup as follows: First, we train the ViT to map the coarse least-square reconstruction $\hat{\mathbf{x}}_{\mathrm{LS}} = \mathbf{F}^{\mathsf{H}}\mathbf{P}^{\mathsf{H}}\mathbf{y}$ to the ground-truth image $\mathbf{x}^* \in \mathbb{C}^n$, which yields an estimate $f_\theta(\hat{\mathbf{x}}_{\mathrm{LS}})$, where $f_\theta$ is the neural network with parameters $\theta$. Note that we use the larger full-sized complex-valued images as opposed to the cropped magnitude images as in Sec. 4. Then, at inference, we replace the Fourier coefficients of our estimate $f_\theta(\hat{\mathbf{x}}_{\mathrm{LS}})$ with the known k-space measurements $\mathbf{y}$.

For this setup, we experimented with several loss functions. For example, using the L1-loss for separately penalizing the real and imaginary part, or applying the SSIM-loss separately on real and imaginary part. We also tried several combinations of L1-loss and SSIM-loss, such as first applying the L1-loss on complex-values, and then combine this loss value together with the loss value after applying the SSIM-loss to the magnitude image.

We made the following observations from our experiments: We found that applying data consistency as described in this paragraph improves reconstruction performance. However to apply a data-consistency term for MRI, we need to train the network to reconstruct a complex-valued image, and for the loss function we used (SSIM, L2, and L1 losses), training a network to reconstruct a complex-valued image and then mapping it to a real-valued one by taking absolute values performs worse than training a ViT to map the cropped magnitude zero-filled image to the cropped magnitude ground-truth image as in Sec. 4. The benefit of the data consistency step did not offset this loss in performance, thus, in our setup a ViT performed best without relying on a data consistency techniques.

