# OpenReview forum: "Vision Transformers Enable Fast and Robust Accelerated MRI"
_MIDL.io/2022/Conference — MIDL 2022_

### Official Review · Reviewer_sv2J · 2022-01-17

**Confidence:** 5
**Preliminary Rating:** 4
**Recommendation:** Oral, Poster

**Summary:**

This paper applied a vision transformer model to MR image reconstruction task and showed performance on-par with U-Net when enough training data were given and better performance than U-Net when pre-trained on ImageNet dataset and fine-tuned on small MR dataset. The proposed ViT MR reconstruction even beat state-of-the-art VarNet when test on another anatomy.

**Strengths:**

There are two strengths in the paper: first, this paper brings freshness into the MR image reconstruction community using deep learning as it is one of the first work applying ViT to MR reconstruction problems; second, the appendix of this paper discussed training details such as training behaviors of U-Net and ViT and performance change with different patch sizes, which are quite informative for readers working on deep learning MR reconstruction and interested in applying ViT to their work.

**Weaknesses:**

I have a few comments:

1. The GitHub link is expired. Please update.
2. How many fine-tuning/training data were used for PT-ViT-L/E2E-VarNet inTable 2 and Fig. 4? Please provide.
3. Following comment 2, it is interesting to see how different training/fine-tuning data affect the anatomy shift performance of PT-ViT-L and VarNet.
4. If possible, it will be very interesting to see how combining VarNet with ViT (e.g. changing the denoising module of VarNet to a ViT-based denoiser) can push the current state-of-the-art MR recon performance.

**Deanonymize Review:**

no

**Paper Type:**

validation/application paper

**Questions To Address In The Rebuttal:**

1. Please provide an updated Github link for code sharing.
2. Please add experiments for different training/fine-tuning data sizes affecting anatomy shift performance.
3. If you can add experiments combining ViT with VarNet, I'd consider raising my final rating.

**Special Issue:**

no

---

### Official Review · Reviewer_d6Fp · 2022-01-23

**Confidence:** 5
**Preliminary Rating:** 5
**Recommendation:** Oral

**Summary:**

The paper investigates the use of Vision Transformers (ViT) to reconstruct undersampled (i.e., accelerated) magnetic resonance imaging (MRI) acquisition. The authors compare their proposal against the traditional U-net model and the End-to-End Variational network using the fastMRI knee and brain datasets. The authors report superior results to the U-net and End-to-End Variational Network when pre-training the ViT using ImageNet.


**Strengths:**

- The paper is well written and the experiments are well designed.
- ViTs are relatively unexplored in the field of MRI reconstruction, making this paper extremely interesting.
- The results are very encouraging.


**Weaknesses:**

- The authors report the results in the validation set of the fastMRI datasets. This is okay, but if the authors used the test sets, their results would be directly comparable to all the other entries in the fastMRI public leaderboard (i.e, the current state-of-the-art).

- Using the root-sum-of-squares of the multi-coil zero-filled reconstructions as the input is probably not optimal. This prevents the addition of data consistency to the proposed model.

- It seems the authors are reporting results in the validation set and not a proper holdout set.

**Deanonymize Review:**

no

**Detailed Comments:**

- The GitHub link for the authors' code seems to be broken.

- In the results tables, the meaning of the emboldened and underlined metrics is not defined in the captions.

- As future work, it would be interesting to consider including data consistency in the ViT model.

**Paper Type:**

methodological development

**Questions To Address In The Rebuttal:**

- The GitHub link for the authors' code seems to be broken. Will the authors make the code publicly available?

- In the results tables, the meaning of the emboldened metrics and underlined metrics are not defined in the captions. Could the authors add this information to the tables' captions?

- It seems the authors are reporting results in the validation set and not a proper holdout set. Could the authors confirm and justify this?

**Special Issue:**

yes

---

### Official Review · Reviewer_1gad · 2022-01-23

**Confidence:** 5
**Preliminary Rating:** 3

**Summary:**

the authors propose a simple yet effective pre-training, which solely relies on big natural image datasets and only 100 MRI training images.  Even when only fine-tuned on 100 MRI images, a pre-trained Vision Transformer yields sharp and detailed reconstructions, showing that the ViT is very suitable for low-data regimes in accelerated MRI.

**Strengths:**

The author propose a simple yet effective pre-training, which solely relies on big natural image datasets, such as ImageNet. We show that pre-training the Vision Transformer drastically improves training data efficiency for accelerated MRI, and increases robustness towards anatomy shifts. In the regime where only 100 MRI training images are available, the pre-trained Vision Transformer achieves significantly better image quality than pretrained convolutional networks and the current state-of-the-art.  Ans the code is available .

**Weaknesses:**

1. The author has mentioned that “Motivated by potential accuracy gains and computational savings, we study Vision Transformers for accelerated magnetic resonance image reconstruction”. However, as far as I know, Transformer does not save computing power but increases it.
2. Besides, the results exceeding UNet cannot prove the effectiveness of this method, because UNet tends to smooth the image in the process of reconstruction, as can be seen from the results in Fig. 4 and 6. The authors should use classical reconstruction algorithms as a baseline, such as MoDL and the recently proposed Complex reel-based DONet (DONet: Dual-Octave Network for Fast MR Image Reconstruction (IEEE Transactions on Neural Networks and Learning Systems)), which are designed for the reconstruction of MRI data.
3. What are the inductive biases that the authors have mentioned in the introduction?
4. The authors should reclarify their statement that “it is yet unclear whether, or to what extent, such results are applicable to the arguably more different realm of accelerated MRI,” because there are many works focused on the MRI reconstruction. Please see an incomplete list here,
-Task Transformer Network for Joint MRI Reconstruction and Super-Resolution (MICCAI 2021)
- Accelerated Multi-Modal MR Imaging with Transformers, https://arxiv.org/abs/2106.14248
-Unsupervised MRI Reconstruction via Zero-Shot Learned Adversarial Transformers https://arxiv.org/abs/2105.08059
5. Training data is no longer scarce thanks to the release of fastMRI. Therefore, the author should reorganize their motivation. It is also unclear whether the target and auxiliary images provided to the network inputs and outputs were complex or magnitude? Did the network have separate channels for representing complex numbers?




**Deanonymize Review:**

yes

**Detailed Comments:**

1. The author has mentioned that “Motivated by potential accuracy gains and computational savings, we study Vision Transformers for accelerated magnetic resonance image reconstruction”. However, as far as I know, Transformer does not save computing power but increases it.
2. Besides, the results exceeding UNet cannot prove the effectiveness of this method, because UNet tends to smooth the image in the process of reconstruction, as can be seen from the results in Fig. 4 and 6. The authors should use classical reconstruction algorithms as a baseline, such as MoDL and the recently proposed Complex reel-based DONet (DONet: Dual-Octave Network for Fast MR Image Reconstruction (IEEE Transactions on Neural Networks and Learning Systems)), which are designed for the reconstruction of MRI data.
3. What are the inductive biases that the authors have mentioned in the introduction?
4. The authors should reclarify their statement that “it is yet unclear whether, or to what extent, such results are applicable to the arguably more different realm of accelerated MRI,” because there are many works focused on the MRI reconstruction. Please see an incomplete list here,
-Task Transformer Network for Joint MRI Reconstruction and Super-Resolution (MICCAI 2021)
- Accelerated Multi-Modal MR Imaging with Transformers, https://arxiv.org/abs/2106.14248
-Unsupervised MRI Reconstruction via Zero-Shot Learned Adversarial Transformers https://arxiv.org/abs/2105.08059
5. Training data is no longer scarce thanks to the release of fastMRI. Therefore, the author should reorganize their motivation. It is also unclear whether the target and auxiliary images provided to the network inputs and outputs were complex or magnitude? Did the network have separate channels for representing complex numbers?




**Paper Type:**

both

**Questions To Address In The Rebuttal:**

1. The author has mentioned that “Motivated by potential accuracy gains and computational savings, we study Vision Transformers for accelerated magnetic resonance image reconstruction”. However, as far as I know, Transformer does not save computing power but increases it.
2. Besides, the results exceeding UNet cannot prove the effectiveness of this method, because UNet tends to smooth the image in the process of reconstruction, as can be seen from the results in Fig. 4 and 6. The authors should use classical reconstruction algorithms as a baseline, such as MoDL and the recently proposed Complex reel-based DONet (DONet: Dual-Octave Network for Fast MR Image Reconstruction (IEEE Transactions on Neural Networks and Learning Systems)), which are designed for the reconstruction of MRI data.
3. What are the inductive biases that the authors have mentioned in the introduction?
4. The authors should reclarify their statement that “it is yet unclear whether, or to what extent, such results are applicable to the arguably more different realm of accelerated MRI,” because there are many works focused on the MRI reconstruction. Please see an incomplete list here,
-Task Transformer Network for Joint MRI Reconstruction and Super-Resolution (MICCAI 2021)
- Accelerated Multi-Modal MR Imaging with Transformers, https://arxiv.org/abs/2106.14248
-Unsupervised MRI Reconstruction via Zero-Shot Learned Adversarial Transformers https://arxiv.org/abs/2105.08059
5. Training data is no longer scarce thanks to the release of fastMRI. Therefore, the author should reorganize their motivation. It is also unclear whether the target and auxiliary images provided to the network inputs and outputs were complex or magnitude? Did the network have separate channels for representing complex numbers?




**Special Issue:**

no

---

### Official Review · Reviewer_E2Q4 · 2022-01-24

**Confidence:** 4
**Preliminary Rating:** 3
**Recommendation:** Poster

**Summary:**

This work presents a vision transformer (ViT) model for fast MRI reconstruction. The original ViT architecture is modified (from a classifier) to have a reconstruction head comprising layer norm and linear layers. Experiments on the fastMRI dataset with reconstruction of knee and brain MR data show the proposed transformer method fare comparably with U-net baseline.

**Strengths:**


* This work explores a pure ViT model for MRI reconstruction which is novel.
* Experiments on two anatomical structures (knee, brain) are presented from the fastMRI challenge dataset, with comparable results compared to U-net baselines.
* Experiments checking domain shifts show ViT models are similar in performance with VarNet.
* Pre-training ViT model shows improved gains compared to pre-training U-net baseline models.
* The method claims improved throughput and lower-memory consumption, which can be beneficial in reconstruction tasks.

**Weaknesses:**

* **Performance in Table 1**: The ViT models reported in Table 1 do not show a large improvment even compared to the simplest of the U-net models (taking the standard deviation into consideration). Perhaps reporting the number of parameters of U-net models could also help contextualize these results.

* **Memory consumption**: In Introduction it is claimed that the ViT models consume less memory compared to baselines. Is this measured as the maximum batch size? If so, that is also higher for U-net-S. Were there any other experiments performed to validate this claim? Also, why do the ViT models consume less memory? Elaborating this could actually strengthen the paper.

* **Performance compared to VarNet**: Why is VarNet not reported in Table 1? And in Table 2, clearly VarNet performs better than ViT-L. Also, pre-training only one of the models gives an advantage to the specific model.


* Is the difference in input reason the VarNet outperforms the ViT model? Why can this information not be used for the ViT models? This is not very clear in the discussion presented in Page 6 where authors say:
> Contrary, the End-to-end VarNet, uses information about the MRI forward model (1)
throughout the network. >



**Deanonymize Review:**

no

**Detailed Comments:**

See points above.

**Final Rating After The Rebuttal:**

4: Weak Accept

**Justification Of The Final Rating:**

Appreciate the thorough responses from the authors. Most of my concerns are satisfactorily addressed.

However, some claims are not entirely justifiable. For instance this statement about MRI images having similar characteristics as natural images might not be valid, as except for some basic morphological properties (edges, texture) there might not be other useful commonalities.
> A reason why we thought pre-training on natural images could work is that natural images have similar characteristics to MRI images.

Nevertheless, the authors have addressed most other concerns satisfactorily, and I am willing to change my score to Weak Accept.

**Paper Type:**

methodological development

**Questions To Address In The Rebuttal:**

* The overall performance improvement of the proposed method is not remarkable, even compared to simple U-net baselines. So the motivation for this work has to be better justified. Why would a ViT model be better than VarNet or U-net? If it is that a pretrained ViT can obtain good performance with 100 scans, why is it so? Also, why is this pretraining not performed on VarNet?
* Elaboration on why ViT models show reduction in memory consumption and implications of this.
* Description on how are the different ViT, U-net models are obtained?

**Special Issue:**

no

---

### Meta-Review · Area_Chair_VhND · 2022-02-20

**Recommendation:** Accept (Poster)
**Confidence:** 5

**Metareview:**

After rebuttal, the manuscript receives 2 weak accept, 1 strong accept, and 1 borderline. The authors have clearly clarified the issues raised by the reviewers.
The only comments are adding more baselines and double-checking the results in figures.
Overall, most reviewers are satisfied with the response given by the authors and are glad to see that the quality of the paper has been improved substantially. It reaches the minimum requirement for publication.

---

### Decision · Program_Chairs · 2022-02-28

Accept